# Viral Conjunctivitis

**DOI:** 10.3390/v15030676

**Published:** 2023-03-04

**Authors:** Tetsuaya Muto, Shinichiro Imaizumi, Koju Kamoi

**Affiliations:** 1Department of Ophthalmology, Dokkyo Medical University Saitama Medical Center, Koshigaya 343-8555, Japan; 2Imaizumi Eye Hospital, Koriyama 963-8877, Japan; 3Department of Ophthalmology and Visual Science, Tokyo Medical Dental University, Tokyo 113-8519, Japan

**Keywords:** conjunctivitis, adenovirus, enterovirus, herpes virus

## Abstract

Viruses account for 80% of all cases of acute conjunctivitis and adenovirus; enterovirus and herpes virus are the common causative agents. In general, viral conjunctivitis spreads easily. Therefore, to control the spread, it is crucial to quickly diagnose illnesses, strictly implement hand washing laws, and sanitize surfaces. Swelling of the lid margin and ciliary injection are subjective symptoms, and eye discharge is frequently serofibrinous. Preauricular lymph node swelling can occasionally occur. Approximately 80% of cases of viral conjunctivitis are caused by adenoviruses. Adenoviral conjunctivitis may become a big global concern and may cause a pandemic. Diagnosis of herpes simplex viral conjunctivitis is crucial for using corticosteroid eye solution as a treatment for adenovirus conjunctivitis. Although specific treatments are not always accessible, early diagnosis of viral conjunctivitis may help to alleviate short-term symptoms and avoid long-term consequences.

## 1. Introduction

Acute conjunctivitis is one of the most common ophthalmic symptoms in the emergency department. Although acute conjunctivitis can be caused by bacteria, allergies, viruses, or parasites, approximately 80% of cases of acute conjunctivitis are caused by viruses. Wearing contact lenses is a known risk factor for microbial keratitis [1]; however, no current evidence suggests that wearing contact lenses is an independent risk factor for developing viral conjunctivitis [2].

Viral conjunctivitis includes epidemic keratoconjunctivitis (EKC) and pharyngoconjunctival fever (PCF) caused by adenovirus, acute hemorrhage conjunctivitis (AHC) caused by enterovirus and coxsackievirus, and herpetic conjunctivitis caused by herpes simplex virus (HSV). Varicella–zoster virus (VZV) [3], measles virus [4], and mumps virus [5,6] also cause conjunctivitis; however, the clinical findings associated with these viruses are slightly different from those associated with other viruses causing viral conjunctivitis. Viral conjunctivitis is predominantly of the follicular type, and characteristics of viral conjunctivitis include redness, blood vessel engorgement, ocular discharge, pain, photophobia, and pseudo membranes. Notably, viral conjunctivitis may appear as EKC in clinical findings, and it is occasionally known as a clinical EKC. EKC infectivity is particularly robust, and EKC infection is prevalent in summer. In Japan, numerous individuals infected with EKC are placed under observation each year. Although most forms of viral conjunctivitis heal naturally [7], patients may sometimes experience deterioration in their quality of life through symptoms such as blurred vision and loss of visual acuity [8,9]. Specific treatment is required to improve quality of life, reduce subjective symptoms, and shorten the treatment period.

In the present review, we discuss the pathogenesis and management of viral conjunctivitis, which involves various symptoms.

## 2. Epidemic Keratoconjunctivitis (EKC)

EKC is a form of acute conjunctivitis caused by adenovirus, and an estimated one million people are known to contract this disease annually in Japan [10]. Rowe et al. first isolated adenoviruses from tissue cultures of human adenoids in 1953 [11]. Adenoviruses are medium-sized (70–100 nm), non-enveloped viruses with an icosahedral nucleocapsid containing a double-stranded linear DNA genome comprising 34–36 kbp [12]. The adenovirus has an icosahedral shape, and fibers extend from its peak (penton base) [12]. All triangular faces except for the penton base are formed through the arrangement of hexon protein groups [13]. Adenoviruses can cause infectious gastroenteritis [12,14], cystitis [12], myocarditis [12], meningoencephalitis [12], and pneumonia [15]. They are categorized into seven species from A to G [16]. Species D adenoviruses predominantly cause EKC [17]. Although adenovirus types 8, 37, 53, 54, 56, 64 [18], and 85 [19] are causative agents, EKC is most commonly caused by type 8 adenovirus [20]. In particular, infection with type 54 is exclusively observed in Japan, and these cases of EKC are considered severe [21,22,23]. A new variety of adenovirus will eventually be discovered through genetic recombination because strains of adenovirus have been isolated and distinguished from healthy individuals.

Adenovirus is highly contagious and may occasionally spread in classrooms [21] or offices. This virus is usually spread through the use of contaminated ophthalmic instruments and eye solutions, hand-to-eye contact with infected personnel, swimming pools, or fomites in close-contact situations. Restriction of clinical activities, which include delaying eye surgery, releasing hospital inpatients early, and closing ophthalmology wards, may be necessary owing to the risk of hospital-acquired EKC infections, which may result in major epidemics in ophthalmology wards [24]. Ward closure can severely affect hospital administration. The risk of transmission is increased because adenovirus can persist for a particularly long period on environmental surfaces. O’Brien et al. reported that it can remain infectious on surfaces for up to 4–5 weeks [25]. It is stable in the presence of many physical and chemical agents, as well as adverse pH conditions [26]. For example, adenovirus is resistant to lipid solvents because it lacks lipids within its structure [27]. Infectivity is optimal between pH 6.5 and 7.4; however, the virus can withstand pH ranges between 5.0 and 9.0 [26]. In addition, adenovirus may remain infectious after freezing [28].

EKC occurs more frequently in adults than in children. The incubation period is approximately 10 days [21], and the main ocular feature is acute follicular conjunctivitis in clinical findings. Conjunctivitis is caused when a projecting fiber from the penton adheres to the receptor in the conjunctival epithelium. The virus enters cells and multiplies in the nuclei. The increasing inclusion bodies observed in the nucleus are comprised of viruses. When cells (filled with inclusion bodies) die, they release the virus. Follicles are formed on the conjunctiva as a result of the immune response of lymphocytes under the conjunctival epithelium [18,29]. The multiplied virus is transported to the preauricular lymph node by lymphatic channels under the conjunctival epithelium [18]. The swelling in the lymph nodes is attributable to the proliferation of antigen-specific lymphocytes [30]. Eye discharge is often watery and frequently accompanied by lacrimation [31]. Conjunctival examination reveals severe hyperemia (Figure 1A,B). Superficial punctate keratopathy, subconjunctival hemorrhage, pseudo membrane, and symblepharon are occasionally observed. If untreated, pseudo membranes incorporate into the host tissue and form scars that may restrict eye movement and/or cause symptoms of dry eye [29]. The contralateral eye is affected in nearly 70% of patients [32]; however, the infection is considerably less severe. After a week, numerous subepithelial infiltration can be visualized on the cornea, and conjunctival inflammation persists for several weeks. Freyler et al. reported that 47% of patients continued to show signs of stromal keratitis 2 years after infection onset [33]. Aoki et al. reported that subepithelial infiltration was observed in 42.6% of the patients in the early stage of infection [24]. Therefore, even in cases of mild or moderate follicular conjunctivitis, subepithelial infiltration should be examined [24]. The surface of the corneal stroma experiences a subepithelial infiltration owing to a late-onset sensitivity reaction to the adenoviral antigen. Although cell infiltration occurs in the cornea, it appears that adenoviral growth does not.

Innumerable lymphocyte pathogens can be observed by microscopic examination of samples collected via conjunctival scraping. In 1996, an immunochromatography rapid diagnosis kit for human adenovirus antigens was introduced [34]. Adenovirus antigen can be identified from the sample by chaffing the conjunctiva with a swab. The specificity is almost 100%, and the sensitivity is approximately 55% [34]. These kits function on the fundamental principle of detecting hexon proteins that comprise the surface of adenoviruses. The positive rate has a deep correlation with total viral load. Clinicians should be particularly vigilant in the initial stages of the infection owing to the considerably low viral load. In such cases, the kit sensitivity becomes relatively low. The effectiveness of a novel adenovirus detection kit that used tears, including conjunctival exudate, was reported subsequently [35]. A unique viral detection tool does not require conjunctival scraping, in contrast to conventional immunochromatography kits that do. Therefore, patients reported less discomfort in the sample collection stage of this novel method. Because the exact viral infection may be determined during a busy outpatient session and the kits have been applied in actual clinical settings, these kits are one of the beneficial culminations of the collective knowledge in the field of ophthalmology virology. These fast diagnosis kits are particularly helpful for reducing wasteful testing for bacteria, herpes, enterovirus, and chlamydia, as well as for preventing infection in the vicinity of patients. In spite of these improvements in diagnosis technology, large-scale hospital or institutional infections are still being reported currently. DNA from viruses can be promptly recognized via a polymerase chain reaction. However, a polymerase chain reaction cannot be performed without special equipment and pertinent technical knowledge. Safak et al. found that adenoviral keratoconjunctivitis could be revived after laser in situ keratomileusis [36]. The conjunctiva may harbor latent adenoviral infection. In reality, latent adenoviral infection develops in the kidney years after the initial infection [37].

## 3. Pharyngoconjunctival Fever (PCF)

PCF is most commonly caused by adenovirus serotype 3 and less commonly by serotypes 2, 4, 7, and 14 [31]. Adenoviruses are classified into seven species from A to G, among which B is the major species in PCF [38].

PCF usually occurs in children [39], and the incubation period varies from 2 to 14 days [40]. Acute follicular conjunctivitis accompanied by upper respiratory tract involvement, localized lymphadenopathy, and fever distinguishes PCF from other diseases. Therefore, patients with PCF often consult pediatrics or otolaryngology clinics. PCF is less severe than EKC in terms of both subjective symptoms and objective findings. The entire process takes one to two weeks. Tonsillar and throat reddening, as well as edema, occur. Fever sometimes reaches about 40 °C [41], and it persists for about a week. Throat and eye findings usually improve considerably by day 7 of the illness, but some constitutional symptoms may persist for two weeks or longer [12]. Because stool contains the virus, it can be an infection source. The incidence of this disease dramatically increases during summers in swimming pools [40]. Water-borne disease outbreaks can be largely controlled by improving water quality; thus, ensuring the hygiene of the swimming pool is of paramount importance. In recent years, clinicians have shown a tendency to associate EKC and PCF with adenoviral conjunctivitis.

Acute follicular conjunctivitis by chlamydia and HSV may appear similar to PCF in clinical findings. Chlamydial conjunctivitis is unilateral, exhibits chronic follicular conjunctivitis, and often accompanies urethritis and trachelitis.

## 4. Acute Hemorrhage Conjunctivitis (AHC)

Ghana experienced a unique conjunctivitis pandemic between June and October of 1969 [42]. Chatterjee et al. named it “epidemic hemorrhagic conjunctivitis”; however, it was locally known as “Apollo 11 disease” because it occurred at approximately the same time as Apollo 11 landed on the moon [43]. Coxsackievirus A-24 variant and enterovirus-70 are the two major causative agents [44], and they are categorized as enterovirus. Enterovirus is one of the main viruses that cause viral conjunctivitis in Japan [45]. The enterovirus has a diameter of ~25 nm [46]. It is characterized by its icosahedral structure and its single positive-strand RNA gene [47]. Today, approximately 7500 base sequences have been elucidated [48]. This virus lacks an envelope and is composed of capsid proteins and nucleic acids. It is susceptible to quick gene changes as it is an RNA virus [49]. Viral reproduction occurs in the host cells [50]. The enterovirus typically multiplies in the intestines and spreads through respiratory secretions, feces, and contaminated water [51]. Joshi et al. reported an association between enterovirus infection and the quality of the water supply [52].

During the 1971 epidemics in Japan, Singapore, and Morocco, enterovirus type 70 was isolated from the conjunctiva of patients with acute hemorrhagic conjunctivitis [53]. Several small-scale epidemics have been reported worldwide since then [54,55,56]. In the 7509 cases reported in Okinawa in 1994, 62% of patients were aged 11–15 years. Junior high school pupils purposefully rubbed their eye discharge together to avoid attending class [56]; nearly 1.5 times as many men as women are affected by AHC [57]. The spread of AHC is most likely to occur after a flood, according to prior research [58]. According to Zhang, there was a distinct seasonal pattern with a peak from August to October and a median onset age of 24 years [59]. AHC is an extremely contagious form of conjunctivitis, and its incubation period is approximately 1 day. A few weeks may pass before an outbreak that started with a few isolated cases or that was transmitted by sick passengers exacerbates into an epidemic. AHC is mainly transmitted through hand-to-eye-to-hand contact [60]. The simultaneous or sequential onset of conjunctivitis in both eyes may occur within a short time after contact with an infectious source [61]. In cases of sudden onset follicular conjunctivitis that are tested negative on an adenovirus detection kit, AHC should be considered, and eyes should be examined. Small-scale outbreaks of AHC are common in primary and secondary schools and as well as colleges in China [62]. One or two students may have originally been infected by the virus during summer vacation and brought the virus back to school, and the disease may have spread promptly owing to the transmission caused by highly frequent contact between students in school [62].

Clinical symptoms can include subconjunctival hemorrhages, punctate epithelial keratitis, punctate lid edema, conjunctival injection, follicular response, and watery discharge [44]. Loss of visual acuity seldom occurs through subepithelial infiltration. AHC symptoms are milder than those of EKC. The most characteristic finding in patients with AHC is subconjunctival hemorrhage, and the disease is named on the basis of this finding. Subconjunctival hemorrhages may appear as small dots or large hemorrhages with a wide range of whole bulbar conjunctiva. The temporal–superior region is the most common site of these hemorrhages.

## 5. Herpes Simplex Virus (HSV) Conjunctivitis

HSV is one of the major viruses that cause a number of infectious disorders worldwide. It is a deoxyribonucleic acid (DNA) virus with two DNA chains [63]. HSV is a member of the alpha subfamily and is made up of an icosahedral capsid encased in an envelope [64]. Although HSV is primarily a human-specific virus, its host range is extensive [65]. Therefore, animal models can be easily created as HSV infects both rats and rabbits [66,67]. Animal models are frequently used to study immunity and latent infection [66,67]. Two types of HSV are known to exist [63]. HSV conjunctivitis is an acute condition that can be caused by an initial or recurring infection [68].

According to estimates, HSV causes 1.3–4.8% of acute conjunctivitis cases [69]. Aside from conjunctivitis, HSV causes keratitis [70,71], anterior uveitis [72,73], and acute retinal necrosis [74,75]. The number of infected individuals increases with increasing age. By the age of 30, at least 50% of Americans are believed to be infected with HSV-1, and by the age of 60, nearly 100% of Americans are thought to have this infection in their trigeminal ganglion [76]. Although type-1 HSV is predominantly observed, type-2 HSV was reported in a few cases [8,77]. Type-2 infects the sexual organs in general. The resulting discharge is thin and watery, and the skin around the eye often develops a bleb with an umbilical fossa and palpebra conjunctivitis with palpebra herpes. Diagnosis is particularly challenging in the absence of skin lesions [68]. According to the national monitoring program for ocular infectious disorders in Japan, HSV was responsible for 4.3% of clinically identified cases of EKC [68].

Recurrent conjunctivitis is unilateral. However, first-time infections are bilateral, and patients may show folliculitis alone and occasionally in combination with superficial punctate keratopathy or dendritic keratitis. Additionally, these symptoms may appear concurrently with preauricular or inframaxillary lymph node enlargement. Bilateral cases of recurrent conjunctivitis are more common than unilateral cases in cases of atopic dermatitis. Kaposi’s varicelliform eruption accompanied by atopic dermatitis is a severe form of this disease during a first-time infection. Smears of conjunctival scraping materials predominantly show lymphocytes. Although HSV conjunctivitis occasionally involves polymorphonuclear leucocytes, adenovirus infection does not. A herpes virus member has a characteristic known as a multinucleated giant cell, and this peculiarity can be used to identify additional viruses. Detection of viral antigen is mostly impossible in cases of conjunctivitis without corneal epithelium lesions and exanthem. Although viral isolation is a time-consuming process, it is possible to establish a diagnosis through this method. Measuring the titer of neutralizing antibodies in partner serum can be useful. Recently, a study reported occasionally utilizing polymerase chain reaction [76].

## 6. Varicella–Zoster Virus (VZV) Conjunctivitis

VZV is a DNA virus that belongs to the alpha subfamily of herpesviridae, and its basic structure is identical to that of HSV. VZV is a human-specific virus, and infection does not occur or is highly restricted in other species [78]. However, HSV is more contagious than other species, including humans. Therefore, designing an experimental model of VZV is challenging, and this issue impedes clinicians’ efforts to elucidate the disease pathogenesis. Primary infection with VZV produces the syndrome of varicella (chickenpox), whereas reactivation of the latent virus in sensory ganglionic neurons results in zoster shingles. Fever, fatigue, stress [79], the use of immunosuppressive drugs [80], and immune depression caused by malignant tumors often reactivate VZV. Furthermore, VZV is known to cause conjunctivitis [3]. In particular, the exanthem around the tip of the nose often accompanies the ocular symptoms. This phenomenon is called “Hutchinson’s sign” [81].

Although a varicella vaccine is available in Japan, its administration is voluntary. Therefore, in most cases, the disease occurs when individuals do not undergo vaccination. This is a reason why the varicella pandemic remains unsolved. Childhood is the most prevalent age for VZV infection, which can spread through contact, droplet, and airborne transmission [82].

Herpes zoster, frequently observed in older adults, is caused by the reactivation of VZV, which may remain dormant in the sensory ganglia after the initial infection in childhood [3]. Most cases of VZV infections in adulthood have the risk of herpes zoster as the antibody retention ratio for VZV is >90%. Approximately 50% of older adults aged >85 years have experienced herpes zoster [83]; VZV is a relatively familiar virus. In addition to moderate catarrhal conjunctivitis in one eye or both eyes, varicella causes the development of a bleb in the vicinity of the palpebra as part of the general bleb genesis. The first division of the trigeminal nerve displays the typical exanthem of herpes zoster (Figure 2A). Furthermore, herpes zoster shows unilateral papilla, follicle, pseudomembrane, and bleb. The complication rate of conjunctivitis is approximately 35.4%, and other eye complications include keratitis (76.2%) and iritis/uveitis (46.6%) [84]. Pressure pain and swelling of the preauricular lymph node are relatively mild. Typically, phlyctenoid findings were observed in the limbic conjunctiva. Conjunctiva appeared injected, edematous, and frequently had petechial hemorrhages (Figure 2B) [85]. The results typically resolve within 1 week. Smear samples from conjunctival scrapings include multinucleated giant cells and mononucleosis. Similarly, a multinucleated giant cell is the defining feature of an HSV infection [86], and this represents a unique phenomenon in viral infection that occurs as a result of cell degeneration.

## 7. Measles Conjunctivitis

Measles virus is an enveloped virus with a single-strand, non-segmented negative-sense RNA genome and exclusively causes disease in old- and new-world non-human primates and humans [87]. The morbillivirus, the causative agent for measles, is a highly contagious illness that predominantly affects children but can potentially infect adults if they have not received the recommended immunization [88]. Once the measles virus is inhaled and a primary target cell is infected, systemic spread ensues and clinical signs appear after 9–19 days [89].

A typical acute febrile measles infection leads to fever, cough, conjunctivitis, coryza, and a distinguishing rash which can lead to complications in multiple organ systems, including pneumonia, otitis media, and encephalitis, with encephalitis being the worst manifestation [90]. Measles is always accompanied by conjunctivitis. Measles-related conjunctivitis is moderate and nonspecific in nature [91]. Because conjunctivitis occurs at the prodromal stage, it is valuable for diagnosis. Koplik patches on the conjunctiva are occasionally accompanied by measles, which also causes catarrhal conjunctivitis. These are clinical findings that indicate the proliferation of the virus under the conjunctiva. Approximately 20% of subconjunctival hemorrhages and 10–20% of follicles can be visualized. Furthermore, measles can cause keratitis [92].

## 8. Mumps Conjunctivitis

The mumps virus is a member of the paramyxoviridae and causes disease mainly in school-aged children and adolescents [93]. Mumps infection is still one of the most common transmissible viral infections, although it is preventable by vaccination [94]. Ocular manifestations of mumps are very rarely seen [5]. Mumps induces various eye symptoms; dacryoadenitis is the most frequent manifestation, indicating that the mumps virus has a tendency to localize in the salivary tissue, including the lacrymal gland [5]. A paramyxovirus is a contagious agent, and it spreads through direct contact via droplets [6]. One-third of those who contract the mumps virus have a subclinical infection, making it a benign and self-limiting illness. Mumps symptoms in the eyes are quite uncommon [5]. In general, conjunctivitis observed in such cases is of the acute follicular type. Mumps rarely induces corneal endothelitis [95] and retrobulbar neuritis [96].

## 9. Treatment and Counterplan

### 9.1. EKC, PCF, and AHC

No particular antiviral drugs have been approved for adenovirus, Coxsackievirus A-24 variant, and enterovirus-70 at present. Many drugs have been tested in preclinical rabbit studies and in clinical trials for the treatment of adenovirus ocular infections. Several have demonstrated antiviral efficacy in preclinical rabbit studies. Perhaps some of these, in addition to ganciclovir, should be highlighted. Symptomatic therapy is the main treatment. Topical antibiotics are used to treat or prevent bacterial superinfection [97]. However, subsequent bacterial infections are extremely uncommon. Owing to issues with antibiotic resistance, preventive antibiotic administration may not be necessary. Steroid eye drops are effective for pseudo membrane and multiple subepithelial infiltrates. Steroid eye drops can remove several subepithelial infiltrates; however, this process may take several months. To treat several subepithelial infiltrates that are resistant to topical steroids, tacrolimus eye drops may be useful [98]. Infants with EKC frequently present with pseudomembranes, and this membrane should be actively removed to prevent symblepharon. Studies on dexamethasone, povidone–iodine, and immune suppression eye drops are being conducted [18,99,100] (Figure 3). Although ganciclovir has been reported to be effective for adenovirus [101], the mechanism underlying its effects remains unclear. Further studies to elucidate this mechanism are warranted.

Owing to the lack of effective disease treatment methods, hygienic measures to prevent EKC, PCF, and AHC outbreaks are very important. Infection may spread via the hands of medical workers and medical instruments. Further, tonometer tips for intraocular pressure measurement and indirect ophthalmoscope lenses, and slit lamps can spread the infection. Multidose eyedrop bottles used by patients with EKC can also act as a vector for viral transmission, which can last for up to 9 weeks [102]; thus, separate eyedrop bottles should be used for the left and right eyes in patients with bilateral eye disease. Hands should be washed under running water and disinfected with ethanol. Repeating this procedure makes it more effective. Disposable paper towels should be used to dry hands after washing. Medical instruments should be washed with water and subsequently submerged in 80% ethanol for more than 10 min. Povidone–iodine is also a useful disinfectant. Disinfection through boiling for 10 s is the best method to sterilize heat-resistant medical instruments because viruses cannot survive in boiling water [103]. The consultation room should be wiped with a disinfectant. Ultraviolet light irradiation is excellent for large consultation rooms.

### 9.2. HSV Conjunctivitis, VZV Conjunctivitis

Although both HSV and VZV can be treated with acyclovir, VZV has a lesser susceptibility than HSV [104]. As a result, acyclovir administration should be increased overall when treating VZV. Thymidine kinase is phosphorylated three times, thereby converting acyclovir into acyclovir triphosphate [105]. Herpes virus DNA polymerase is inhibited by acyclovir triphosphate [106], and it prevents DNA synthesis, in particular by ingesting viral DNA [105]. Acyclovir eye ointment should be painted 5 times/day for conjunctivitis and exanthema. Painted times should be reduced by the disappearance of conjunctival conditions. Topical antibiotics are administered to stop subsequent bacterial infections. A total of 1000 mg of acyclovir or valacyclovir are given intravenously if nonspecific symptoms are present (Figure 3). Topical corticosteroids should be avoided because they potentiate the virus and may cause harm [107]. They occasionally result in stellate keratitis, dendritic keratitis, and necrotizing keratitis. Recently, there have been a few reports of HSV cases that were acyclovir-resistant [76,108,109]. In the case of acyclovir-resistant, amenamevir is efficient [110]. Amenamevir will be subsequently used for ocular disease. Amenamevir does not phosphorylate but inhibits helicase–primase directly [111]. Because the mechanism is quite different from acyclovir, clinicians hope that amenamevir can have further clinical application when compared with acyclovir.

### 9.3. Measles Conjunctivitis

Treatment of uncomplicated measles cases typically involves supportive care, including antipyretics, antitussives, hydration and/or environmental controls (for example, humidification). There are currently no antiviral treatments for measles that are clinically effective [4] (Figure 3).

### 9.4. Mumps Conjunctivitis

Onal et al. reported that topical corticosteroid and cycloplegic, and a hyperosmolar solution were effective for a child with mumps kerato–uveitis [112]. According to Matoba, mumps-associated ocular illness is self-limiting and has no long-term effects [91]. In conclusion, symptomatic therapy plays a major role because no specific treatment has been reported thus far (Figure 3).

## 10. Conclusions

Caution should be exercised because pediatric viral conjunctivitis infections may frequently occur come with pseudomembranes. Treatment for viral conjunctivitis is supportive in general. Adenovirus is responsible for the majority of cases of viral conjunctivitis. Management protocols for adenoviral conjunctivitis have not changed currently. Specific antiviral drugs may change the steps involved in the handling of this disease clinically. Cases of adenoviral conjunctivitis should be remarked on from now on. Herpetic conjunctivitis is treated by acyclovir ointment in general. Clinicians must be vigilant in identifying cases of conjunctivitis with acyclovir resistance. Steroids must be used with care, and a herpetic infection, which may get exacerbated following steroid administration, should be eliminated only after a thorough ophthalmologic examination. Increased viral replication may result from herpetic conjunctivitis treated with corticosteroids.

## Figures and Tables

**Figure 1 viruses-15-00676-f001:**
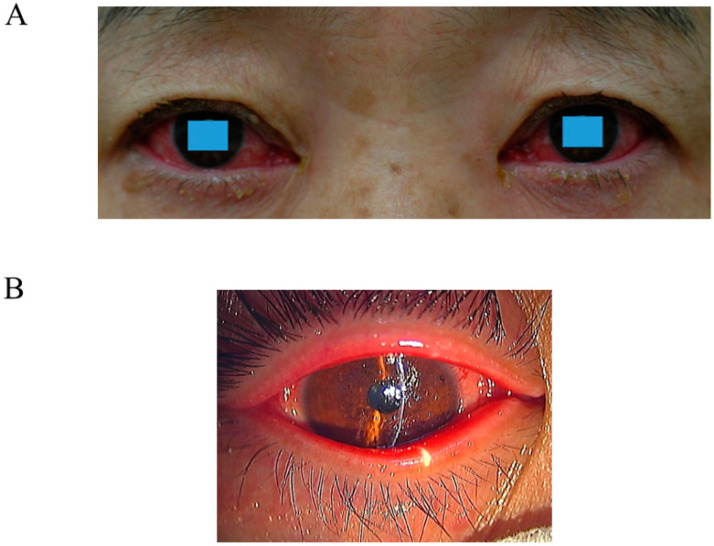
(**A**) Ciliary injection and eye discharges around the eyes are seen in both eyes of a patient with EKC. (**B**) Hard ciliary injection and swelling around the eye are observed in a patient with EKC. These pictures are provided by Dr. Kaoru Araki-Sasaki (Department of Ophthalmology, Kansai Medical University, Hirakata, Japan).

**Figure 2 viruses-15-00676-f002:**
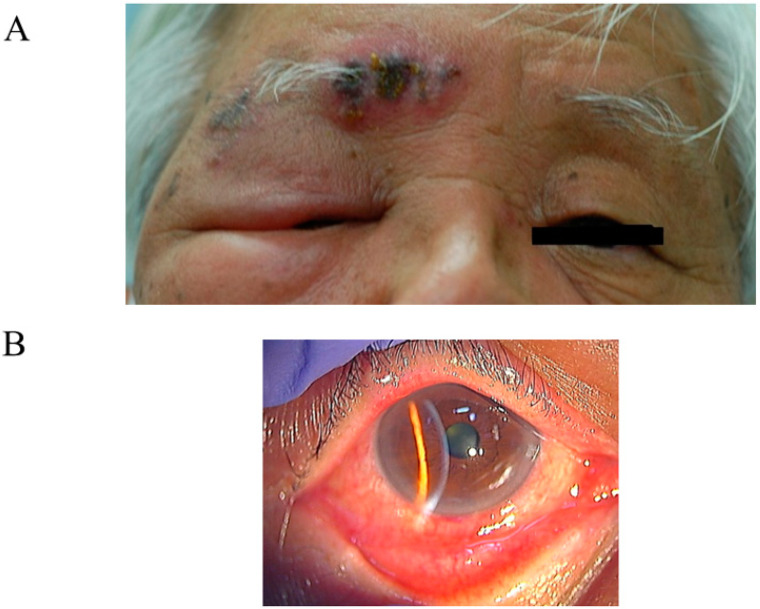
(**A**) Exanthem is observed on the upper eyelid around the eyebrow caused by VZV. (**B**) There is a hard ciliary injection and follicle formation in a patient with VZV conjunctivitis. These pictures are provided by Dr. Kaoru Araki-Sasaki (Department of Ophthalmology, Kansai Medical University, Hirakata, Japan).

**Figure 3 viruses-15-00676-f003:**
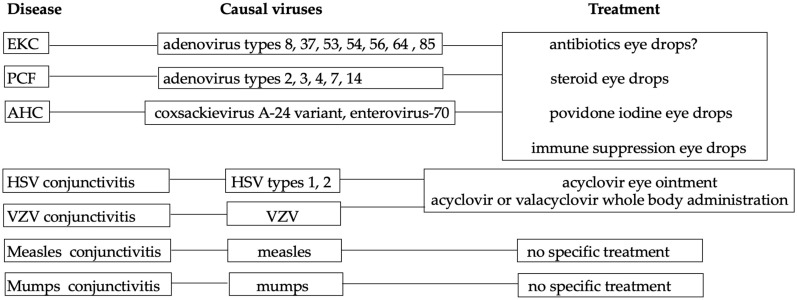
A diagram shows summary of each disease. The diagram provides a comprehensive summary of various diseases presented in this review, including their underlying causes and available treatments.

## Data Availability

No new data were created or analyzed in this study. Data sharing is not applicable to this article.

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
