# Peer review of "Viral Conjunctivitis"

_viruses, 2023, doi:10.3390/v15030676_

Round 1

Reviewer 1 Report

The authors present a review on viral conjunctivitis. Several revisions should be made to strengthen the manuscript.

English editing is needed throughout the manuscript. There are numerous awkward sentences and phrases that must be revised.

The lines of the manuscript must be labelled for ease of review.

In the PCF section, it is stated “Although cleaning the tank for the lower half of the body at the pool is often abolished lately, this disease sometimes comes into fashion dramatically through the pool in summer [26].”. This sentence must be revised to clarify its meaning.

The last sentence in the HSV conjunctivitis must be referenced.

In the VZV conjunctivitis section, it is stated “Herpes zoster also shows unilateral papilla, follicle, pseudo membrane and bleb. The complication rate of conjunctivitis is about 35.4% [41].”. The complications of VZV conjunctivitis must be described since the complication rate is referred to.

Similar to the HSV conjunctivitis section, the last sentence in the VZV conjunctivitis must be referenced.

In the Measles conjunctivitis section, it is stated “The Morbilli virus, which causes measles, is a highly (approximately 95%) contagious illness that mostly affects children but can potentially infect adults if they have not had the recommended immunization [43]. Firstly, the Measles virus is a “morbillivirus” and not a “Morbilli virus”. Secondly, the term “highly (approximately 95%) contagious illness” must be clarified. What does 95% contagious refer to?

In the Measles conjunctivitis section, it is also stated “A typical acute febrile measles infection leads to fever, cough, conjunctivitis, coryza, and a distinguishing rash which can lead to complications in multiple organ systems such as pneumonia, otitis media, and the worst of all—encephalitis.30”. This statement must be referenced with the correct reference.

The legends to all of the figures must be expanded to fully describe the figures so they can stand alone. The causative agent must be included in all figures. The letters “a” and “b” must be capitalized in Figures 1 and 3 to be consistent with the legends.

I cannot find reference #23 in the literature. This reference must be confirmed.

Author Response

Dr.Eric O. Freed

Editor-in-Chief

Viruses

Manuscript ID: viruses-2125114
Type of manuscript: Review
Title: Viral conjunctivitis
Authors: Tetsuya Muto *, Shinichiro Imaizumi, Koju Kamoi

Submitted to section: Human Virology and Viral Diseases,

Dear Editor:

We appreciate the thorough reviews of our submitted manuscript, “Viral conjunctivitis.” The reviewers’ constructive comments were invaluable to improving our manuscript, and we thank them for their time and efforts.

In the following pages, please find a list of our specific revisions and additions to the manuscript.

We sincerely thank you for your comments and hope that you will find that our revised manuscript is now suitable for publication in Viruses.

Yours sincerely,

Tetsuya Muto, MD, PhD,

Department of Ophthalmology, Dokkyo Medical University Saitama Medical Center, 2-1-50 Minamikoshiagya, Koshigaya City, Saitama, 343-8555, Japan.

Tel: +81 48 965 1111

Fax: +81 48 965 1127

E-mail: ueda.castle@gmail.com

Response to the reviewers

Reviewer 1

The authors present a review on viral conjunctivitis. Several revisions should be made to strengthen the manuscript.

English editing is needed throughout the manuscript. There are numerous awkward sentences and phrases that must be revised.

Thank you for your comment. Yes, we asked English editing service again. We have certification document as below.

The lines of the manuscript must be labelled for ease of review.

Thank you for your comment. Yes, we labelled each lines.

In the PCF section, it is stated “Although cleaning the tank for the lower half of the body at the pool is often abolished lately, this disease sometimes comes into fashion dramatically through the pool in summer [26].”. This sentence must be revised to clarify its meaning.

Thank you for your comment. Yes, we revised as below.

The incidence of this disease dramatically increases during summers in swimming pools [32]. Water-borne disease outbreaks can be largely controlled by improving water quality; thus, ensuring the hygiene of swimming pool is of paramount importance.

The last sentence in the HSV conjunctivitis must be referenced.

 Thank you for your comment. Yes, we revised as below.

Recently, a study reported occasionally utilizing polymerase chain reaction [58].

In the VZV conjunctivitis section, it is stated “Herpes zoster also shows unilateral papilla, follicle, pseudo membrane and bleb. The complication rate of conjunctivitis is about 35.4% [41].”. The complications of VZV conjunctivitis must be described since the complication rate is referred to.

Thank you for your comment. Yes, we revised as below.

“Furthermore, herpes zoster shows unilateral papilla, follicle, pseudo membrane, and bleb. The complication rate of conjunctivitis is approximately 35.4% [63] and other eye complications include keratitis (76.2%) and iritis/uveitis (46.6%) [63].”

Similar to the HSV conjunctivitis section, the last sentence in the VZV conjunctivitis must be referenced.

Thank you for your comment. Yes, we revised as below.

 Similarly, a multinucleated giant cell is the defining feature of an HSV infection [65], and this represents a unique phenomenon in viral infection that occurs as a result of cell degeneration.

In the Measles conjunctivitis section, it is stated “The Morbilli virus, which causes measles, is a highly (approximately 95%) contagious illness that mostly affects children but can potentially infect adults if they have not had the recommended immunization [43]. Firstly, the Measles virus is a “morbillivirus” and not a “Morbilli virus”. Secondly, the term “highly (approximately 95%) contagious illness” must be clarified. What does 95% contagious refer to?

Thank you for your comment. Yes, we revised as below.

The morbillivirus, the causative agent for measles, is a highly contagious illness that predominantly affects children but can potentially infect adults if they have not received the recommended immunization [66]. 

In the Measles conjunctivitis section, it is also stated “A typical acute febrile measles infection leads to fever, cough, conjunctivitis, coryza, and a distinguishing rash which can lead to complications in multiple organ systems such as pneumonia, otitis media, and the worst of all—encephalitis.30”. This statement must be referenced with the correct reference.

Thank you for your comment. Yes, we revised as below.

“A typical acute febrile measles infection leads to fever, cough, conjunctivitis, coryza, and a distinguishing rash which can lead to complications in multiple organ systems, including pneumonia, otitis media, and encephalitis, with encephalitis being the worst manifestation [68].”

The legends to all of the figures must be expanded to fully describe the figures so they can stand alone. The causative agent must be included in all figures. The letters “a” and “b” must be capitalized in Figures 1 and 3 to be consistent with the legends.

 Thank you for your comment. Yes, we revised letters from “a” and “b” to “A” and “B”.

I cannot find reference #23 in the literature. This reference must be confirmed.

Thank you for your comment. Yes, this reference is written in Japanese. You can find it below URL.

https://webview.isho.jp/openurl?rft.genre=article&rft.issn=0370-5579&rft.volume=72&rft.issue=4&rft.spage=481

Reviewer 2 Report

The work is not of great interest. The therapy is treated very superficially.

Author Response

Dr.Eric O. Freed

Editor-in-Chief

Viruses

Manuscript ID: viruses-2125114
Type of manuscript: Review
Title: Viral conjunctivitis
Authors: Tetsuya Muto *, Shinichiro Imaizumi, Koju Kamoi

Submitted to section: Human Virology and Viral Diseases,

Dear Editor:

We appreciate the thorough reviews of our submitted manuscript, “Viral conjunctivitis.” The reviewers’ constructive comments were invaluable to improving our manuscript, and we thank them for their time and efforts.

In the following pages, please find a list of our specific revisions and additions to the manuscript.

We sincerely thank you for your comments and hope that you will find that our revised manuscript is now suitable for publication in Viruses.

Yours sincerely,

Tetsuya Muto, MD, PhD,

Department of Ophthalmology, Dokkyo Medical University Saitama Medical Center, 2-1-50 Minamikoshiagya, Koshigaya City, Saitama, 343-8555, Japan.

Tel: +81 48 965 1111

Fax: +81 48 965 1127

E-mail: ueda.castle@gmail.com

Reviewer 3 Report

This article is a review of the viruses that can cause conjunctivitis.  Such a review is considered important in the field.  Before publication, a revision is needed.  The quality of the writing is not consistent and in certain sections needs to be improved: for example, " They can cause conjunctivitis but not many [3]. Although there is a varicella vaccine in Japan, this is voluntary vaccination. As a result, the varicella pandemic does not get better."  Also, acronyms. while defined at the beginning, should not be used as headlines for each section. For example, EKC (adenoviral conjunctivitis) epidemic keratoconjunctivitis by adenovirus should be clearer: e.g, Adenovirus as the cause of epidemic keratoconjunctivitis (EKC). Also, in some cases it seems that facts were gathered and pasted together without clarity and evenness of flow.  Even full accuracy should be strictly adhered to.  For example: . "Recently, there have been a report of HSV conjunctivitis case that were acyclovir-resistant [39]"  does not clarify that this is caused by HSV-2.  No mention of the fact that it is HSV-1 that is the major subtype of herpes that is responsible.  This is important, because HSV-1 from oral cavity cold sores can migrate from the trigeminal ganglia to the eye.  HSV-2 is mainly cervical.

A diagram, depicting the eye and pointing out the pathology would be helpful.   Moreover each section should mention the same parameters, For example frequency of the disease in the global population; treatment if any including drugs.  As such a Summary table would be very helpful.

Author Response

Dr.Eric O. Freed

Editor-in-Chief

Viruses

Manuscript ID: viruses-2125114
Type of manuscript: Review
Title: Viral conjunctivitis
Authors: Tetsuya Muto *, Shinichiro Imaizumi, Koju Kamoi

Submitted to section: Human Virology and Viral Diseases,

Dear Editor:

We appreciate the thorough reviews of our submitted manuscript, “Viral conjunctivitis.” The reviewers’ constructive comments were invaluable to improving our manuscript, and we thank them for their time and efforts.

In the following pages, please find a list of our specific revisions and additions to the manuscript.

We sincerely thank you for your comments and hope that you will find that our revised manuscript is now suitable for publication in Viruses.

Yours sincerely,

Tetsuya Muto, MD, PhD,

Department of Ophthalmology, Dokkyo Medical University Saitama Medical Center, 2-1-50 Minamikoshiagya, Koshigaya City, Saitama, 343-8555, Japan.

Tel: +81 48 965 1111

Fax: +81 48 965 1127

E-mail: ueda.castle@gmail.com

Reviewer 3

This article is a review of the viruses that can cause conjunctivitis.  Such a review is considered important in the field.  Before publication, a revision is needed.  The quality of the writing is not consistent and in certain sections needs to be improved: for example, " They can cause conjunctivitis but not many [3]. Although there is a varicella vaccine in Japan, this is voluntary vaccination. As a result, the varicella pandemic does not get better."  

Yes, we revised as below.

 Furthermore, VZV is known to cause conjunctivitis [3]. In particular, the exanthem around the tip of nose often accompanies the ocular symptoms. This phenomenon is called “Hutchinson’s sign“[61]. Although a varicella vaccine is available in Japan, its administration is voluntary. Therefore, in most cases, the disease occurs when individuals do not undergo vaccination. This is a reason why the varicella pandemic remains unsolved.

Also, acronyms. while defined at the beginning, should not be used as headlines for each section. For example, EKC (adenoviral conjunctivitis) epidemic keratoconjunctivitis by adenovirus should be clearer: e.g, Adenovirus as the cause of epidemic keratoconjunctivitis (EKC).

Yes, we deleted adenoviral conjunctivitis as below.

EKC (adenoviral conjunctivitis)

Also, in some cases it seems that facts were gathered and pasted together without clarity and evenness of flow.  Even full accuracy should be strictly adhered to.  For example: . "Recently, there have been a report of HSV conjunctivitis case that were acyclovir-resistant [39]" does not clarify that this is caused by HSV-2.  No mention of the fact that it is HSV-1 that is the major subtype of herpes that is responsible.  This is important, because HSV-1 from oral cavity cold sores can migrate from the trigeminal ganglia to the eye.  HSV-2 is mainly cervical.

Thank you for your comment. Yes, we revised as below.

Recently, a number of HSV cases have demonstrated acyclovir resistance [59, 82,83]. 

A diagram, depicting the eye and pointing out the pathology would be helpful. Moreover each section should mention the same parameters, For example frequency of the disease in the global population; treatment if any including drugs.  As such a Summary table would be very helpful.

Thank you for your comment. Yes, we made figure 4 anew.

Round 2

Reviewer 1 Report

I commend the authors on the English editing throughout the manuscript. The manuscript reads much better. The additional background information added to the specific virus sections is very useful. However, much of the new background information added has not been referenced. All statements in these passages must be referenced in full. Specifically:

1.       Lines 51-54.

2.       Lines 71-75.

3.       Lines 81-85.

4.       Lines 150-159.

5.       Lines 188-195.

6.       Lines 230-231.

7.       Lines 236-237.

8.       Lines 257-259.

9.       Lines 274-277.

10.   Lines 317-321.

Each of the sections should be divided into multiple paragraphs. For example, the EKC section is one paragraph that is a page and a half long. It is much easier for the readers to read the article when there are multiple paragraphs.

As I mentioned in my initial review, the legends to all of the figures must be expanded to fully describe the figures so they can stand alone. The condition (EKC, Figure 1) and causative agents (VZV, Figures 3A and 3B) must be included in all figures. As written, the reader has no idea what virus is causing the conditions described. Furthermore, since this is a review of viral conjunctivitis, Figure 2 should be deleted as there should not be a photo of HSV-1 keratitis in the article.

What is meant by the term “mononucleosis” on lines 100 and 213. To this reviewer, the term “mononucleosis” refers to infectious mononucleosis caused by Epstein-Barr Virus (EBV)? Does this refer to a monocyte response? Please revise this term to remove any ambiguity.

There are several specific revisions that must be made for clarity:

1.       Lines 32-33: “however, the clinical findings associated with these viruses are slightly different from those associated with THE viruses causing”. Replace the “THE” with “other” or a similar word to differentiate the virus categories.

2.       Lines 55-56: “They are categorized into 7 categories from A to G. Group G is predominantly found in cases of EKC.” Firstly, adenoviruses are categorized into 7 “Species”, not “categories” or “Groups”.

3.       Line 56: “Group G is predominantly found in cases of EKC.” This statement is incorrect. “Species D” adenoviruses predominantly cause EKC.

4.       Lines 127-128: “Adenoviruses are classified to 7 groups from A to G, among which B is the major group in PCF.” This sentence should read “Adenoviruses are classified to 7 Species from A to G, among which B is the major Species in PCF.”.

5.       Lines 136-137: “Because stool contains A virus, it can be an infection source.” Change “A” to “the”.

6.       Lines 149, 287, Figure 4: “Coxsackie” must be revised to “Coxsackievirus”.

7.       Lines 228-230: “Primary infection with VZV produces the syndrome of varicella (chicken pox), whereas reactivation of the latent virus in sensory ganglionic neurons results in zoster.”. Add the more common term “(shingles)” after the last word zoster.

8.       Lines 287-288: “No particular antiviral drugs EXIST for adenovirus, Coxsackie A-24 variant, and enterovirus-70 at present.”. Replace the term “exist” with “have been approved”. Many drugs have been tested in preclinical rabbit studies and in clinical trials for the treatment of adenovirus ocular infections. Several have demonstrated antiviral efficacy in preclinical rabbit studies. Perhaps some of these, in addition to ganciclovir, should be highlighted.

9.       Lines 295-296: “Infant frequently accompanies pseudo membrane and this membrane should be actively removed to prevent symblepharon.”. This sentence if very awkward. A possible revision could be: “Infants with EKC frequently present with pseudomembranes and this membrane should be actively removed to prevent symblepharon.”.

10.   Lines 303-304: “Further, SYNOPHTHALMIA tips for intraocular pressure measurement and SYNOPHTHALMIA lenses, and slit-lamps can spread the infection.”. The term “synophthalmia” is a synonym of “cyclopia” a genetic disorder in humans and animals. I’m sure the authors do not mean that. A possible revision could be “Further, tonometer tips for intraocular pressure measurement and indirect ophthalmoscope lenses and slit-lamps can spread the infection.”.

11.   Lines 311-313: “Disinfection through boiling for 10 seconds is the best method to sterilize heat-resistant medical instruments because viruses cannot survive in boiling water [80].”. However, on line 75 it is stated “In addition, adenovirus is heat resistant and may remain infectious after freezing.”. Is adenovirus really heat resistant? These points should be qualified.

12.   Lines 317-318:  Although both HSV and VZV can be treated with acyclovir, VZV has a lesser sensitivity than HSV.”. Replace “sensitivity” with “susceptibility”.

13.   Line 325: Valtrex must be capitalized as it is a brand name, or substitute valacyclovir for Valtrex.

14.   Line 357-358: “Caution should be exercised because baby viral conjunctivitis infections may frequently occur come with pseudo membranes.”. Replace “baby” with “pediatric”. Pseudomembranes should be one word.

Author Response

Comments and Suggestions for Authors

I commend the authors on the English editing throughout the manuscript. The manuscript reads much better. The additional background information added to the specific virus sections is very useful. However, much of the new background information added has not been referenced. All statements in these passages must be referenced in full. Specifically:

  1. Lines 51-54.

Thank you for your comment. We revised as below.

The adenovirus has an icosahedral shape, and fibers extend from its peak (penton base) [12]. All triangular faces except for the penton base are formed through the arrangement of hexon protein groups [13]. Adenoviruses can cause infectious gastroenteritis [12, 14], cystitis [12], myocarditis [12], meningoencephalitis [12], and pneumonia [15]. They are categorized into 7 species from A to G [16]. Species D, B and E adenoviruses predominantly cause EKC [17].

  1. Lines 71-75.

Thank you for your comment. We revised as below.

It is stable in the presence of many physical and chemical agents, as well as adverse pH conditions [26]. For example, adenovirus is resistant to lipid solvents because it lacks lipids within its structure [27]. Infectivity is optimal between pH6.5 and 7.4; however, the virus can withstand pH ranges between 5.0 and 9.0 [26]. In addition, adenovirus may remain infectious after freezing [28].

  1. Lines 81-85.

Thank you for your comment. We revised as below.

Follicles are formed on the conjunctiva as a result of the immune response of lymphocytes under the conjunctival epithelium [18, 29]. The multiplied virus is transported to the preauricular lymph node by lymphatic channels under the conjunctival epithelium [18]. The swelling in the lymph nodes is attributetable to the proliferation of antigen-specific lymphocytes [30].

  1. Lines 150-159.

Thank you for your comment. We revised as below.

Enterovirus is one of the main viruses that cause viral conjunctivitis in Japan [45]. The enterovirus has a diameter of ~25 nm [46]. It is characterized by icosahedral structure and its single positive-strand RNA gene [47]. Today, the approximately 7500 base sequences have been elucidated [48]. This virus lacks an envelope and is composed of capsid proteins and nucleic acids. It is susceptible to quick gene changes as it is an RNA virus [49]. Viral reproduction occurs in the host cells [50]. The enterovirus typically multiplies in the intestines and spreads through respiratory secretions, feces, and contaminated water [51]. Joshi et al. reported an association between enterovirus infection and the quality of the water supply [52].

  1. Lines 188-195.

Thank you for your comment. We revised as below.

It is a deoxyribonucleic acid (DNA) virus with two DNA chains [63]. HSV is a member of the alpha subfamily and is made up of an icosahedral capsid encased in an envelope [64]. Although HSV is primarily a human-specific virus, its host range is extensive [65]. Therefore, animal models can be easily created as HSV infects both rats and rabbits [66, 67]. Animal models are frequently used to study immunity and latent infection [66,67]. Two types of HSV are known to exist [63]. HSV conjunctivitis is an acute condition that can be caused by the initial or recurring infection [68].

  1. Lines 230-231.

Thank you for your comment. We revised as below.

Fever, fatigue, stress [79], the use of immunosuppressive drugs [80], and immune depression caused by malignant tumors often reactivate VZV.

  1. Lines 236-237.

Thank you for your comment. We revised as below.

Childhood is the most prevalent age for VZV infection, which can spread through contact, droplet, and airborne transmission [82].

  1. Lines 257-259.

Measles virus is an enveloped virus with a single strand, non-segmented negative sense RNA genome and exclusively causes disease in old- and new-world non-human primates and humans [87].

  1. Lines 274-277.

The mumps virus is a member of the paramyxoviridae and causes disease mainly in school-aged children and adolescents [93]. Mumps infection is still one of the most common transmissible viral infections although it is preventable by vaccination [94]. Ocular manifestations of mumps are very rarely seen [95].

  1. Lines 317-321.

 Although both HSV and VZV can be treated with acyclovir, VZV has a lesser sensitivity than HSV [105]. As a result, acyclovir administration should be increased overall when treating VZV. Thymidine kinase is phosphorylated three times, thereby converting acyclovir into acyclovir triphosphate [106]. Herpes virus DNA polymerase is inhibited by acyclovir triphosphate [107], and it prevents DNA synthesis in particular by ingesting viral DNA [106].

Each of the sections should be divided into multiple paragraphs. For example, the EKC section is one paragraph that is a page and a half long. It is much easier for the readers to read the article when there are multiple paragraphs.

Thank you for your comment. We divided 4 paragraphs in EKC section, 3 paragraphs in PCF

section, 3 paragraphs in AHC section, 3 paragraphs in HSV conjunctivitis section, 3 paragraphs in

VZV conjunctivitis section, 2 paragraphs in Measles conjunctivitis section.

As I mentioned in my initial review, the legends to all of the figures must be expanded to fully describe the figures so they can stand alone. The condition (EKC, Figure 1) and causative agents (VZV, Figures 3A and 3B) must be included in all figures. As written, the reader has no idea what virus is causing the conditions described. Furthermore, since this is a review of viral conjunctivitis, Figure 2 should be deleted as there should not be a photo of HSV-1 keratitis in the article.

Thank you for your comment. We revised figure legends as below.

Figure 1. (A) Ciliary injection and eye discharges around the eyes are seen in both eyes of a patient with EKC. (B) Hard ciliary injection and swelling around the eye are observed in a patient with EKC. These pictures are provided by Dr. Kaoru Araki-Sasaki (Department of Ophthalmology, Kansai Medical University, Hirakata, Japan).

Figure 2. (A) Exanthem is observed on the upper eyelid around the eyebrow caused by VZV. (B) There is a hard ciliary injection and follicle formation in a patient with VZV conjunctivitis. These pictures are provided by Dr. Kaoru Araki-Sasaki (Department of Ophthalmology, Kansai Medical University, Hirakata, Japan).

Figure 3. A diagram shows summary of each disease. The diagram provides a comprehensive summary of various diseases presented in this review, including their underlying causes and available treatments.

We deleted Figure 2 (a photo of HSV-1 keratitis).

What is meant by the term “mononucleosis” on lines 100 and 213. To this reviewer, the term “mononucleosis” refers to infectious mononucleosis caused by Epstein-Barr Virus (EBV)? Does this refer to a monocyte response? Please revise this term to remove any ambiguity.

Thank you for your comment. We changed the word “mononucleosis” to “lymphocyte” to remove any ambiguity.

There are several specific revisions that must be made for clarity:

  1. Lines 32-33: “however, the clinical findings associated with these viruses are slightly different from those associated with THE viruses causing”. Replace the “THE” with “other” or a similar word to differentiate the virus categories.

Thank you for your comment. We revised as below.

however, the clinical findings associated with these viruses are slightly different from those

associated with other viruses causing viral conjunctivitis.

  1. Lines 55-56: “They are categorized into 7 categories from A to G. Group G is predominantly found in cases of EKC.” Firstly, adenoviruses are categorized into 7 “Species”, not “categories” or “Groups”.

Thank you for your comment. We revised as below.

“They are categorized into 7 species from A to G [16]. Species D adenoviruses predominantly

cause EKC [17].”

  1. Line 56: “Group G is predominantly found in cases of EKC.” This statement is incorrect. “Species D” adenoviruses predominantly cause EKC.

Thank you for your comment. We revised as below.

“Species D adenoviruses predominantly cause EKC.”

  1. Lines 127-128: “Adenoviruses are classified to 7 groups from A to G, among which B is the major group in PCF.” This sentence should read “Adenoviruses are classified to 7 Species from A to G, among which B is the major Species in PCF.”.

Thank you for your comment. We revised as below.

“Adenoviruses are classified to 7 species from A to G, among which B is the major species in PCF [38].”

  1. Lines 136-137: “Because stool contains A virus, it can be an infection source.” Change “A” to “the”.

Thank you for your comment. We revised as below.

“Because stool contains the virus, it can be an infection source.”

  1. Lines 149, 287, Figure 4: “Coxsackie” must be revised to “Coxsackievirus”.

Thank you for your comment. We revised as below.

Coxsackievirus A-24 variant and enterovirus-70 are the two major causative agents [36],

No particular antiviral drugs exist for adenovirus, Coxsackievirus A-24 variant, and

enterovirus-70 at present.

Figure 3

  1. Lines 228-230: “Primary infection with VZV produces the syndrome of varicella (chicken pox), whereas reactivation of the latent virus in sensory ganglionic neurons results in zoster.”. Add the more common term “(shingles)” after the last word zoster.

Thank you for your comment. We revised as below.

“Primary infection with VZV produces the syndrome of varicella (chicken pox), whereas reactivation

of the latent virus in sensory ganglionic neurons results in zoster shingles.”

  1. Lines 287-288: “No particular antiviral drugs EXIST for adenovirus, Coxsackie A-24 variant, and enterovirus-70 at present.”. Replace the term “exist” with “have been approved”. Many drugs have been tested in preclinical rabbit studies and in clinical trials for the treatment of adenovirus ocular infections. Several have demonstrated antiviral efficacy in preclinical rabbit studies. Perhaps some of these, in addition to ganciclovir, should be highlighted.

Thank you for your comment. We revised as below.

“No particular antiviral drugs have been approved for adenovirus, Coxsackievirus A-24 variant, and

enterovirus-70 at present. Many drugs have been tested in preclinical rabbit studies and in clinical

trials for the treatment of adenovirus ocular infections. Several have demonstrated antiviral efficacy

in preclinical rabbit studies. Perhaps some of these, in addition to ganciclovir, should be

highlighted.”

  1. Lines 295-296: “Infant frequently accompanies pseudo membrane and this membrane should be actively removed to prevent symblepharon.”. This sentence if very awkward (ぎこちない). A possible revision could be: “Infants with EKC frequently present with pseudomembranes and this membrane should be actively removed to prevent symblepharon.”.

Thank you for your comment. We revised as below.

“Infants with EKC frequently present with pseudomembranes and this membrane should be actively

removed to prevent symblepharon.”

  1. Lines 303-304: “Further, SYNOPHTHALMIA tips for intraocular pressure measurement and SYNOPHTHALMIA lenses, and slit-lamps can spread the infection.”. The term “synophthalmia” is a synonym of “cyclopia” a genetic disorder in humans and animals. I’m sure the authors do not mean that. A possible revision could be “Further, tonometer tips for intraocular pressure measurement and indirect ophthalmoscope lenses and slit-lamps can spread the infection.”.

Thank you for your comment. We revised as below.

“Further, tonometer tips for intraocular pressure measurement and indirect ophthalmoscope lenses

and slit-lamps can spread the infection.”

  1. Lines 311-313: “Disinfection through boiling for 10 seconds is the best method to sterilize heat-resistant medical instruments because viruses cannot survive in boiling water [80].”. However, on line 75 it is stated “In addition, adenovirus is heat resistant and may remain infectious after freezing.”. Is adenovirus really heat resistant? These points should be qualified.

Thank you for your comment. We revised on line 75 as below.

“In addition, adenovirus may remain infectious after freezing.”

  1. Lines 317-318:  “Although both HSV and VZV can be treated with acyclovir, VZV has a lesser sensitivity than HSV.”. Replace “sensitivity” with “susceptibility”.

Thank you for your comment. We revised as below.

“Although both HSV and VZV can be treated with acyclovir, VZV has a lesser susceptibility than

HSV.”

  1. Line 325: Valtrex must be capitalized as it is a brand name, or substitute valacyclovir for Valtrex.

Thank you for your comment. We revised as below.

A total of 1000 mg of acyclovir or valacyclovir are given intravenously if nonspecific symptoms

are present (Fig 3).

  1. Line 357-358: “Caution should be exercised because baby viral conjunctivitis infections may frequently occur come with pseudo membranes.”. Replace “baby” with “pediatric”. Pseudomembranes should be one word.

Thank you for your comment. We revised as below.

“Caution should be exercised because pediatric viral conjunctivitis infections may frequently occur

come with pseudomembranes.”

Reviewer 3 Report

This edited version is acceptable for publication.  The text has been greatly improved.

Author Response

Thank you for review.